SciPost Physics

# General Relativity and the Ricci Flow

Mohammed Alzain[1*]

**1** Department of Physics, Omdurman Islamic University, Khartoum, Sudan
* malzain1992@gmail.com

August 10, 2021

## Abstract

In Riemannian geometry, the Ricci flow is the analogue of heat diffusion; a deformation of the metric tensor driven by its Ricci curvature. As a step towards resolving the problem of time in quantum gravity, we attempt to merge the Ricci flow equation with the Hamilton-Jacobi equation for general relativity.

## 1 Introduction

The quantum theory and general relativity are based on two different and incompatible notions of time, and this represents a serious obstacle on the way to unify these two theories within a single framework [1]. In quantum theory, as in classical physics, time is a fixed background parameter used to mark the evolution of the system; it is not a physical observable that can be described by an operator. In fact, a key ingredient in the time-dependent Schrödinger equation is the background Newtonian time. In contrast, time in general relativity is a coordinate on the spacetime manifold whereas the spacetime is a dynamical object -namely the spacetime in general relativity is not merely a background for physical processes but there is a reaction between the geometry of spacetime and its matter-energy content. As a corollary, the Newtonian picture of time evolution does not exist in standard general relativity.

Despite the fact that the concept of an external time parameter is incompatible with general relativity because it is a diffeomorphism-invariant theory, general relativity could still be forced into a Newtonian framework, without breaking diffeomorphism-invariance,

by allowing the spacetime manifold to undergo a geometric flow known as the Ricci flow, since the Ricci flow is invariant under the full diffeomorphism group.

## 2  The Ricci Flow

The Ricci flow equation [2]

$$\frac{\partial g_{\mu\nu}}{\partial \lambda} = -R_{\mu\nu}, \tag{1}$$

describes the deformation of the Riemannian metric $g_{\mu\nu}$ with respect to an auxiliary time variable $\lambda$, where $R_{\mu\nu}$ is the Ricci curvature tensor. It was introduced by Hamilton [2] to smooth out the geometry of the manifold to make it look more symmetric. The analogy with the heat equation becomes apparent in harmonic coordinates (i.e. coordinates which are harmonic functions for the metric) [3], since in these coordinates the Ricci curvature equals $-\frac{1}{2}$ the Laplacian of the metric, up to lower order terms. Hence, in the same way that the heat equation spreads the temperature evenly throughout space, the Ricci flow spreads the curvature evenly throughout space.

Hamilton also proved that the evolution equation (1) has a unique solution for a short time for any given smooth metric on a closed manifold. As an illustrative example of a Ricci flow, suppose the initial metric is an Einstein metric (it satisfies the vacuum Einsten's field equations with a non-zero cosmological constant) and has positive Ricci curvature then the manifold evolves by shrinking and eventually collapsing to a point in finite time [4]. It is then clear that under the Ricci flow singularities will frequently develop in finite time. Hamilton proposed a method to continue the flow after singularities by performing topological-geometric surgeries shortly before the singular time [5].

## 3  The Einstein-Ricci Flow

To develop an extension of Einstein's general relativity that incorporates the Ricci flow, Graf [6] proposed that the Einstein field equations should be upgraded from the hyperbolic form

$$R_{\mu\nu} - \frac{1}{2}Rg_{\mu\nu} = \kappa T_{\mu\nu}, \tag{2}$$

to the parabolic form

$$R_{\mu\nu} - \frac{1}{2}Rg_{\mu\nu} + \frac{\partial g_{\mu\nu}}{\partial \lambda} = \kappa T_{\mu\nu}. \tag{3}$$

In this sense, metrics that are solutions of the Einstein field equations are fixed points of this flow, with the energy-momentum tensor $T_{\mu\nu}$ playing the role of a source term. Unfortunately, from the perspective of a partial differential equation, this flow behaves badly. As noted by Hamilton [2], the equation (3) cannot have solutions, even for a short time, because it implies a backward heat equation in the Ricci scalar $R$.

There is another possibility; we may substitute (1) directly into Einstein's field equations. Thus we obtain,

$$\frac{\partial g_{\mu\nu}}{\partial \lambda} + \frac{1}{2}Rg_{\mu\nu} = -\kappa T_{\mu\nu}, \tag{4}$$

This is the unnormalized Yamabe flow [7] with a source term. However, this flow does not correlate with the full Einstein's field equations; we observe that in the case of no flow, $\partial g_{\mu\nu}/\partial\lambda = 0$, the energy-momentum conservation, $\nabla^\mu T_{\mu\nu} = 0$, cannot be recovered.

We then arrive at the conclusion that the attempt to merge the Einstein field equations with the Ricci flow into a heat equation (an Einstein-Ricci flow) is not tenable. Hence, we

seek a different route to achieve this unification. The starting point is the Hamiltonian formulation of general relativity; the so-called ADM formalism [8].

The canonical, or Hamiltonian, formulation of general relativity requires a 3+1 dimensional decomposition of spacetime such that the spacetime manifold is foliated into a set of spacelike hypersurfaces of constant time, and each hypersurface is equipped with a three-dimensional metric $g_{ij}$ which will play the role of the configuration variable in the canonical formalism. The momenta $\pi_{ij}$ canonically conjugate to $g_{ij}$ are expressible as linear combinations of the extrinsic curvature of the hypersurface. The Hamiltonian for classical general relativity is a sum of two constraints on the initial values of the canonically conjugate variables $g_{ij}$ and $\pi_{ij}$, and these constraints are known as the Hamiltonian constraint

$$(\frac{1}{2}g_{ij}g_{kl} - g_{ik}g_{jl})\pi^{ij}\pi^{kl} + gR = 0, \tag{5}$$

and the momentum constraints

$$\nabla_j \pi^{ij} = 0, \tag{6}$$

where $R$ is the three-dimensional scalar curvature and $g = det g_{ij}$.

Peres [9] elucidated how the introduction of an arbitrary scalar functional of the three-dimensional metric $S(g)$, with the only requirement that the functional form of $S$ remains invariant under coordinate transformations, implies that the momentum constraints (6) are automatically satisfied, given that

$$\pi^{ij} = \frac{\delta S}{\delta g_{ij}}. \tag{7}$$

The functional $S$ can therefore be identified as the Hamilton-Jacobi functional for the gravitational field and the differential equation that results from substituting (7) into (5) is the Hamilton-Jacobi equation for general relativity,

$$(\frac{1}{2}g_{ij}g_{kl} - g_{ik}g_{jl})\frac{\delta S}{\delta g_{ij}}\frac{\delta S}{\delta g_{kl}} + gR = 0. \tag{8}$$

As demonstrated in [10] this single equation is fully equivalent to all ten components of the vacuum Einstein's field equations.

At this stage, a proper integration of the Ricci flow with general relativity might be possible if we replaced the tensor equation (1) with a scalar equation as follows: instead of evolving the metric by its Ricci tensor, we suggest to evolve a scalar functional of the metric $S(g)$ by the scalar curvature of the same metric;

$$\frac{\partial S(g)}{\partial \lambda} = -R, \tag{9}$$

where $\lambda$ is an external time parameter and $R$ is the Ricci scalar curvature, this is a generalized and a more complicated version of the Ricci flow (1) but the central idea is still preserved; which is the evolution of metrics by their Ricci curvature.

The flow (9) can be defined on any Riemannian manifold of arbitrary dimension once the metrics $g$ are given and a functional $S(g)$ is constructed out of them, but we are only interested in its relevance to the canonical formulation of general relativity. Let us assume that $S(g)$ obeys the Hamilton-Jacobi equation (8). This, then, leads to a new Hamilton-Jacobi equation where the functional $S$ contains an explicit time dependence,

$$(\frac{1}{2}g_{ij}g_{kl} - g_{ik}g_{jl})\frac{\delta S}{\delta g_{ij}}\frac{\delta S}{\delta g_{kl}} - g\frac{\partial S}{\partial \lambda} = 0. \tag{10}$$

The analogy between (10) and the Hamilton-Jacobi equation in classical mechanics [11] indicates that a separation of variables can be effected in order to obtain a solution; where the time dependence in $S$ can be separated off by making the ansatz:

$$S = S_1(g) + S_2(\lambda). \tag{11}$$

Consequently, the left hand side of (10) will depend only on the configuration variable $g$, while the right hand side will depend only on the time $\lambda$. Then, both sides cannot be equal unless they are equal to a common constant $\beta$; hence,

$$g^{-1}(\frac{1}{2}g_{ij}g_{kl} - g_{ik}g_{jl})\frac{\delta S_1(g)}{\delta g_{ij}}\frac{\delta S_1(g)}{\delta g_{kl}} = \frac{\partial S_2(\lambda)}{\partial \lambda} = \beta. \tag{12}$$

The Hamilton-Jacobi equation can then be split into two equations, an equation for $S_1(g)$,

$$g^{-1}(\frac{1}{2}g_{ij}g_{kl} - g_{ik}g_{jl})\frac{\delta S_1(g)}{\delta g_{ij}}\frac{\delta S_1(g)}{\delta g_{kl}} = \beta, \tag{13}$$

and the other for $S_2(\lambda)$,

$$\frac{\partial S_2(\lambda)}{\partial \lambda} = \beta, \tag{14}$$

where $\beta$ is the separation constant, thus, the solution $S$ can be obtained in accordance with the ansatz (11).

## 4  Conclusion

We conclude that the deformation of Riemannian metrics by their Ricci curvature introduces the Newtonian notion of time evolution into the structure of general relativity by allowing the spacetime manifold to evolve (even in the absence of matter) with respect to an external time variable, hence, bridging a major gap between the general theory of relativity and the quantum theory.

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
