# Peer review of "General Relativity and the Ricci Flow"

_SciPost Physics_

## Round 1 · Referee Report · Anonymous · 2021-9-9

Weaknesses

The paper requires more careful work given the aim of discovering new physics.

Report

I cannot recommend the paper for publication given its ambitious claim. If one takes Equations (9) and (14) and (11) once cannot reconcile the results. R is a function of $g_{\mu \nu}$ and from the Ansatz (11) I can't see why (9) would be true, unless R is a constant, and that is a very restrictive set of geometries.

Requested changes

The author reformulate the action using a more careful analysis.

  • validity: poor
  • significance: ok
  • originality: ok
  • clarity: good
  • formatting: good
  • grammar: good

Author:  Mohammed Alzain  on 2021-09-11

(in reply to Report 1 on 2021-09-09)
Category:
reply to objection

The observation made by the referee that R must be a constant for the ansatz (11) to be true is accurate, this is indeed a very restrictive set of geometries but it must be noted that it has been proposed for the sole purpose of finding a solution. The constancy of R is a necessary requirement for the validity of the ansatz (11) but not a necessary requirement for the validity of the equation (9). Equation (9) remains valid without the requirement that R must be a constant unless the ansatz (11) is imposed.

---

## Round 1 · Referee Report · Anonymous · 2021-9-12

Report

In this manuscript, the author first discusses the Ricci flow which is an evolution equation for the metric of a smooth manifold driven by its Ricci curvature. This flow has been used successfully to prove the Poincare conjecture in mathematics.
The author then describes several attempts to incorporate the Ricci flow in physics by modifying the Einstein equation. The modifications are totally ad hoc and the author then outlines several problems with these modifications. The author then discusses the Hamiton-Jacobi functional for Einstein gravity, and introduces in a completely ad hoc manner, eqn (9) in the manuscript which is the evolution of this functional under a new flow parameter \lambda, the flow being driven by the scalar curvature. This is then put back in the Hamilton-Jacobi equation. This whole procedure is poorly motivated, and this ad hoc modification of the Hamilton-Jacobi equation is not even analyzed to work out how the resulting theory differs from Einstein gravity and what are its consequences. This is unsuitable for publication in SciPost Physics.

---

## Editorial Decision

editor-in-charge_assigned